# Exploring the benefits of full-time hospital facility dogs working with nurse handlers in a children's hospital

**Natsuko Murata-Kobayashi**[1]*, **Keiko Suzuki**[1,2], **Yuko Morita**[1], **Harumi Minobe**[2], **Atsushi Mizumoto**[3], **Shiro Seto**[2]

**1** Specified Nonprofit Organization Shine On Kids, Chuo-ku, Tokyo, Japan, **2** Shizuoka Children's Hospital, Shizuoka, Japan, **3** Kansai University, Suita-shi, Osaka, Japan

* natsuko@sokids.org

**Data Availability Statement:** All data and R code used in this study are available in the Open Science Framework (https://osf.io/hxktc/).

## Abstract

### Objective

To examine the benefits of full-time hospital facility dogs (HFDs) working with qualified nurse handlers for inpatients in a pediatric medical facility.

### Methods

A questionnaire survey on the evaluation of HFD activities was conducted in a hospital that had introduced HFDs for the first time in Japan and has been using them for 9 years. Of the 626 full-time medical staff, 431 responded, of which 270 who observed HFD activities were included in the analysis. The Questionnaire contained 20 questions, and nine questions were selected for presentation in this paper because they focused on the situations in which HFD activities were thought to have a strong impact on inpatients. A comparison of the respondents' evaluations for each question was made, and differences in the respondents' attributes (such as profession, length of clinical experience and experience of dog owner-ship) for those items were examined.

### Results

The impact of HFDs in terminal care was ranked highest among the respondents. Similarly, HFDs increased patient cooperation for clinical procedures. The responses to these two items did not differ statistically depending on the respondents' attributes. The results imply that patients were more cooperative even for highly invasive examinations and procedures with the support of HFD activities.

### Conclusions

Healthcare providers considered that HFDs were useful, especially for providing support during the terminal phase and for gaining patients' cooperation for procedures. The fact that the handler was a nurse and the HFD team worked full-time may have enhanced the effectiveness of the program.

**Funding:** The authors received no specific funding for this work.

**Competing interests:** The authors have declared that no competing interests exist.

## Background

Facility Dogs (FDs) are professionally-trained dogs that work alongside health or human service professionals to address client/patient specific goals within the scope of practice of the handler [1]. In particular, FDs working in hospitals are called hospital facility dogs (HFDs). They are increasingly being used as a form of psychosocial care for patients with childhood cancer and other serious illnesses. HFDs differ from therapy dogs, which are trained pets [1], in that they are as professionally trained as guide dogs and service dogs. HFDs also usually live with their handlers as caretakers and commute with them every morning. Although there are no official statistics, the world's largest training organization trains 83 new FDs per year [2]. In Japan, the first HFD team started working in 2010 at Shizuoka Children's Hospital [3], in collaboration with the non-profit organization, Shine On! Kids. In recent years, an increasing number of medical professionals are serving as HFD handlers, including clinical psychologists [4], child life specialists [5], occupational therapists, and physical therapists. The program investigated in this study is unique as it employed nurses as handlers and operated on a full-time basis.

The use of conventional therapy dogs has spread because of their low cost derived from volunteer services [6]. Compared to volunteer therapy dogs programs, there are higher costs associated with launching and implementing a HFD team in a pediatric hospital. In the case of our program, in addition to the labor cost of the nurses who serve as handlers, there is the cost of training a HFD to service dog standards. In the U.S. training a service dog is estimated to cost $10,000 [7]. In Japan the total cost for training and lifetime maintenance of a service dog is 3.88–5.39 million yen [8]. Given the significant financial investment, it's important to examine the impact and value of the HFD team's activities. Additionally, animal welfare must be addressed through the creation of a system to optimize and prioritize the working time and activities of a HFD team [9]. Most hospitals in Japan have only one HFD team, even large hospitals with hundreds of beds, making it impossible for all pediatric patients to receive support. In order to develop and expand sustainable HFD activities in these hospitals, it is vital to determine the priority of interventions.

In previous studies of volunteer therapy dog activities, various patient changes have been reported after a single intervention of about 20 min [10–13], but the long-term effects have been inconsistent [14, 15]. In contrast, interventions by medical professional handlers have been reported to be positive [4, 16, 17], but these interventions were conducted on a short-term basis in comparison to our full-time HFDs and medical professional handlers. Therefore, to the best of our knowledge, no previous studies have extensively examined the effectiveness of different types of interventions under the following set of conditions: the handler is a medical professional, the dog is a HFD, and both the handler and HFD work full time (i.e., not temporarily). In other words, the interventions that the medical personnel find the most useful under the combination of these three conditions are yet to be identified.

A review of previous studies has revealed that the intervention research design has room for improvement. For example, in a review on animal-assisted interventions (AAIs) for cancer and palliative care patients [18], while some positive benefits were reported, only one of the ten studies reviewed was a randomized controlled trial (RCT). Indeed, further studies with a more rigid research design are needed to obtain reliable evidence on the effectiveness of the interventions.

Interventions conducted by nurses serving as handlers may range from less invasive medication assistance to highly invasive procedures and treatments conducted in collaboration with other professionals [19]. For this reason, in the current study, it was first necessary to

develop a questionnaire that could assess the benefits of full-time HFD activities for inpatients in a pediatric medical facility.

By administering the questionnaire, we aimed to examine and describe the effects of full-time HFD activities and interventions on inpatients in a pediatric medical facility in Japan. The results of this study may help prioritizing the type of interventions that should be conducted by a HFD with a nurse-handler, as well as evaluating the social return on investment (SROI) of such a program. Moreover, this study was a necessary first step for future studies with an RCT design to collect data at a high level of evidence.

## Methods and materials

### Design

This study was an opt-out anonymous questionnaire survey of hospital medical staff and was approved by the Ethics Committee of Shizuoka Children's Hospital (2019–11). Consent was waived because the research does not involve the acquisition of new samples or personal information.

### Participants

The survey was conducted at Shizuoka Children's Hospital between July and September 2019. Questionnaire forms were distributed to all 626 full-time medical staff members. To ensure the anonymity of the respondents, an ID was assigned to them. The analysis was conducted by the first author of this article, who is not affiliated with the hospital.

### Setting

The study was conducted in a 279-bed public hospital where HFD-handler teams had been active since January, 2010: (a) Bailey (Golden Retriever, male, neutered, active for 2.5 years from January 2010–June 2012) and Yuko Morita (nurse) and (b) Yogi (Golden Retriever, male, neutered, active for 9 years from July 2012 to September 2021) and Keiko Suzuki (nurse). Each HFD–handler pair (Yogi and Suzuki during the study period) worked full time at the hospital. Both dogs were trained and certified as HFDs by Assistance Dogs of Hawaii (https://www.assistancedogshawaii.org/), an organization accredited by Assistance Dogs International (ADI), and the two nurses had completed specialized training as handlers. We followed and complied with the published ethical guidelines of ADI and IAHAIO (International Association of Human-Animal Interaction Organizations) [20] by limiting the time of HFD activities, providing the dogs with regular care by a veterinarian, handling them based on positive reinforcement, and ensuring regular visits by a qualified dog trainer.

Interventions were conducted daily on weekdays, with each session lasting approximately 45 min. Two or three, 45-minute intervention sessions were conducted daily on weekdays. A total of 5–20 patients were visited by the team per day. The inclusion criterion for eligible patients was absence of fear or allergy to dogs or symptoms of infection. The number of cases where patients did not visit due to their fear or allergies to dogs was minimal, around 10% or less. All the activities were limited to the wards. The handler was allowed by the hospital to borrow a Personal Handy-phone System (a low-powered wireless phone used within a hospital), access and record data from electronic medical records, and participate in palliative care conferences. The decision regarding interventions with individual patients was made based on the information obtained in conferences and assessments by the medical staff. Thereafter, the handlers scheduled the order and frequency of interventions.

## Questionnaire development

The questionnaire consisted of two sections: one on the demographics of the respondents and the other on the evaluation of HFD activities. The demographics were: (a) type of medical profession, (b) ward the respondent belonged to, (c) years of clinical experience, and (d) whether or not the staff member had ever owned a dog. Because the years of clinical experience affect palliative care knowledge [21], this study was also categorized into three subgroups using quartiles in an exploratory manner to explore the impact of clinical experience. It should be noted that because items Q1-Q10 are not directly related to the study objectives, we provided information on these items exclusively in the online supplementary material.

Since the activities involving nurse-handlers included a wide range of interventions, the items evaluating HFD activities were developed according to previous research and handlers' experiences. We first asked a question about the changes observed in the patients (Q11: Patient cooperation) and staff (Q12: Improvement in workload) without specifying the situations. The changes observed in specific situations were asked in an open-ended format for each item, and the respondents answered in writing. Items 14 to 16 were based on previous research and inquired about: Q14, Reduced medication [14, 22]; Q15, Effects on terminal care [23]; Q16, Effects on expression [24]. Additionally, we asked about situations that the handlers had experienced in the past while working with other professionals: Q13, Support for patient decision making; Q17, Flexibility to schedule change; Q18, Reduction in verbal abuse and violence; and Q19, Ease of outpatient care and readmission (Table 1). All items were rated on a 5-point Likert scale ranging from 1 to 5 (1: Never, 2: Not very often, 3: Sometimes, 4: Very often, 5: Always). The Cronbach alpha reliability coefficient for this part of the questionnaire (Q11–Q19) was .90, indicating a high degree of reliability and consistency within the item set.

## Data analysis

All data analyses were performed using R version 4.0.3 [25]. Comparisons between the type of medical profession, clinical experience, and dog ownership experience were performed using the Kruskal-Wallis test. The effect size ($r$) was calculated along with 95% confidence intervals (CI). Multiple comparisons were performed using the pairwise Mann-Whitney's U test, and the Bonferroni method was employed to adjust the significance level in multiple comparisons. All data and R code used in this study are available in the Open Science Framework (https://osf.io/hxktc/).

# Results

## Participant characteristics

Of the 626 staff members, 431 (69%) completed the questionnaire (Fig 1). Of them, 161 staff members with a response of 1 (never) to Q8, "Have you ever accompanied a facility dog when visiting a patient or performing an examination or procedure?," were excluded to avoid respondent bias [26]. The remaining 270 staff members were included in the analysis (Table 2).

Their profession breakdown was as follows: 33 physicians (12%), 198 nurses (73%), and 39 other professions (14%). The nurses belonged to 14 different wards or departments. Among them, pediatric cancer patients were mainly admitted to two wards, and 36 nurses (18%) belonged to these two wards. The mean number of years of clinical experience at the time of completion of the survey was 14.68 years (standard deviation 26 = 9.96), with 52 (19%). The median number of years of clinical experience was 11.00 years (quartile range 5.50–20.00), respondents having experience of less than 5 years, 144 (53%) between 5 and 20 years, and 70

**Table 1. Twenty question items related to hospital facility dogs.**

| No. | | Question |
|---|---|---|
| 1 | Impact on patients[†‡] | Do you think the facility dog program will have a positive impact on patients? |
| 2 | Impact on families[†‡] | Do you think the facility dog program has a positive impact on the guardians of children? |
| 3 | Impact on staff[†‡] | Do you think the facility dog program will have a positive impact on the staff? |
| 4 | Risk of transmission of infectious diseases[†] | Have you ever felt that facility dogs can be a medium of infection? |
| 5 | Risk of biting[†] | Have you ever feared that a facility dog might bite a person? |
| 6 | Impression of dogs[†] | Has your impression of dogs changed since you learned about facility dogs? |
| 7 | Frequency of meeting | Have you ever met a facility dog in the hospital? |
| 8 | Frequency of accompaniment to interventions | Have you ever accompanied a facility dog when visiting a patient or performing an examination or procedure? |
| 9 | Frequency of requesting interventions | Have you ever requested a facility dog to visit a patient or accompany him/her to an examination or procedure? |
| 10 | Burden on the workload | Have you ever felt that the intervention of a facility dog was a burden on your workload? |
| 11 | Patient cooperation | Hove you ever felt that the intervention of the facility dog helped you to gain the cooperation of a patient for a treatment or procedure? |
| 12 | Improvement in workload[‡] | Have you ever felt that having a facility dog to attend a child during a procedure or examination facilitated your own work? |
| 13 | Support for patient decision making | Have you ever felt that the intervention of the facility dog made it easier for the patient to make decisions in situations where you had to inform him/her of the name of the disease or the prognosis (e.g., announcement of cancer, about surgery, and termination of active treatment)? |
| 14 | Reduced medication[‡] | Have you experienced a decrease in the degree or frequency of drug use during examinations, procedures, or surgeries due to the intervention of the facility dog? |
| 15 | Effects on terminal care[‡] | Have you found the intervention of a facility dog to be useful in palliative care in the terminal phase? |
| 16 | Effects on expression | Have you ever felt that the child's expression of became more active after the intervention of the facility dog? |
| 17 | Flexibility to schedule change | Have you ever felt that the facility dog's intervention helped you in dealing with a child who had difficulty in accepting a change in the schedule or situation? |
| 18 | Reduction in verbal abuse and violence | When dealing with a children or family members who verbally abuse, are violent, use negative words, or have negative attitudes, have you ever felt that their behavior improved when you performed the intervention with a facility dog? |
| 19 | Ease of outpatient care and readmissions | Have you ever felt that the presence of a facility dog in your hospital has made discharged patients less reluctant to attend the outpatient clinic or to be readmitted? |
| 20 | Motivation for joining the hospital and reasons for continuing employment[†] | Is the facility dog a reason why you decided to work at the hospital or why you want to continue working there? |

These responses identified the frequency of each behavior on a scale of 1 to 5, with 5 being the highest level.

[†]For these items, responses indicated the degree of impact of the facility dogs, with 5 being the highest intensity.

[‡]Questions with an open-ended (free text) response.

(26%) more than 20 years, while 4 (4%) did not respond. Furthermore, 130 respondents (48%) had dog ownership experience, 129 (48%) did not have dog ownership experience, and 11 (4%) did not answer.

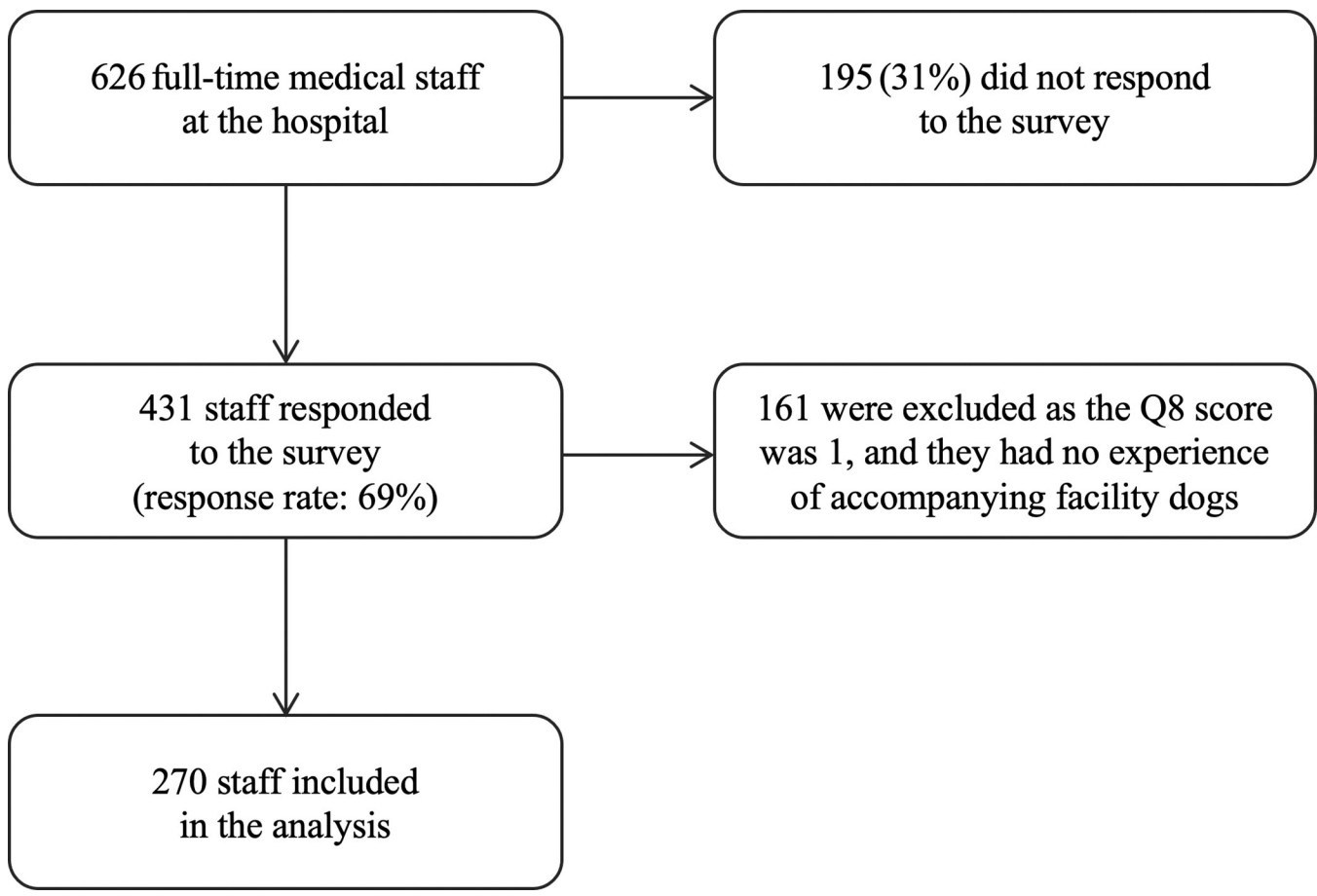

**Fig 1. From July to August 2019, there were 626 full-time medical staff members working in the hospital, and 431 responded to the survey.** Of them, 161 were excluded because their score for Q8 was 1 and they had no experience in accompanying hospital facility dog interventions.

### Examination of the nine items (Q11–Q19) related to the evaluation of HFD activities

The stacked bar graph for each question is shown in Fig 2, and the comparison of the results by respondent attributes is shown in Table 3.

The item with the highest combined percentage for the responses "Very often" and "Always" was Q15, "Impact on terminal care." Of the 270 respondents, 77 (29%) answered this item, of whom 22 (29%) answered "Always," 34 (44%) answered "Very often," 18 (23%) answered "Sometimes," 3 (4%) answered "Not very often," and none answered "Never." A total of 73% respondents answered "Always" and "Very often." There was no statistical difference in the profession, clinical experience, or dog ownership experience among the respondents of this item.

The next highest-rated item was Q11, "Patient cooperation," which refers to the ease of obtaining cooperation from patients for treatment and procedures (Table 1). The number of responses to this item was 193 (72%), the highest among the nine items (Q11–Q19). Of the respondents, 48 (25%) answered "Always," 92 (48%) answered "Very often," 40 (21%) answered "Sometimes," 12 (6%) answered "Not very often," and one (1%) answered "Never." The total number of respondents who answered "Always" and "Very often" reached 73%. This item also did not show any statistical difference in terms of the profession, clinical experience, or dog ownership experience among the respondents.

**Table 2. Demographic information of the respondents and analysis subjects.**

| Demographics | Number of Respondents | (%) | Number of subjects analyzed | (%) |
|---|---|---|---|---|
| Profession | 431 | | 270 | |
| Doctor | 49 | 11 | 33 | 12 |
| Nurse | 308 | **71** | 198 | **73** |
| Others | 74 | 17 | 39 | 14 |
| Ward the nurse belongs to | 308 | | 198 | |
| Neonatal Intensive Care Unit (NICU), Growing Care Unit (GCU) | 51 | **17** | 24 | **12** |
| Internal medicine infant ward (age <3 years) | 20 | 6 | 14 | 7 |
| Infection observation ward | 20 | 6 | 12 | 6 |
| Internal medicine children's ward (age >3 years) and school-age children's ward | 23 | 7 | 22 | 11 |
| Perinatal Centre | 33 | 11 | 24 | **12** |
| Cardiovascular Ward | 10 | 3 | 2 | 1 |
| Cardiac Care Unit (CCU) | 32 | 10 | 22 | 11 |
| Surgical Department | 28 | 9 | 19 | 10 |
| Pediatric Intensive Care Unit (PICU) | 23 | 7 | 19 | 10 |
| Psychological Treatment for Children and Family | 17 | 6 | 13 | 7 |
| Outpatient | 20 | 6 | 9 | 5 |
| Operating rooms | 14 | 5 | 9 | 5 |
| Regional Medical Liaison Office | 7 | 2 | 2 | 1 |
| Nurse management office | 10 | 3 | 7 | 4 |
| Clinical experience | 431 | | 270 | |
| <5 years | 106 | 25 | 52 | 19 |
| 5–20 years | 213 | **49** | 144 | **53** |
| >20 years | 104 | 24 | 70 | 26 |
| Unknown | 8 | 2 | 4 | 1 |
| Experience of having a dog | 413 | | 259 | |
| Yes | 199 | 46 | 130 | 48 |
| No | 214 | **50** | 129 | 48 |
| Unknown | 18 | 4 | 11 | 4 |

Boldface indicates the largest proportion of responses.

The reason for the low number of responses could be related to the frequency of the interventions assumed in each item. For example, regarding "Impact on terminal care" (Q15), the number of deaths discharged from the same hospital is 27 per year (2019) [28]. Therefore, even if HFD intervened in all 27 cases, the number of staff on site would be very limited. On the other hand, situations in which "Patient cooperation" (Q11) occur frequently on a daily basis, and there are many opportunities to be present in such situations. As a result, we obtained responses from 193 respondents (71%).

The item with the lowest positive impact was Q14, "Reduced medication," including sedatives for examinations and procedures. The combined percentage of respondents who answered "Always" and "Very often" was 24%, while that of those who answered "Not very often" and "Never" was 33%, making this item the only one among the nine items where the ratios of the combined percentages were reversed. Differences by respondent demographics were also observed. Respondents with a longer clinical experience tended to score higher: the three groups were significantly different ($\chi^2$ = 13.93, $df$ = 2, $p < 0.001$), with a moderate effect size of 0.33 (95% CI: 0.14, 0.49) (See details in Table 3). Respondents with dog ownership experience also tended to score higher than those with no experience ($\chi^2$ = 4.15, $df$ = 1, $p = 0.042$) with a small effect size of 0.20 (95% CI: 0.01, 0.38).

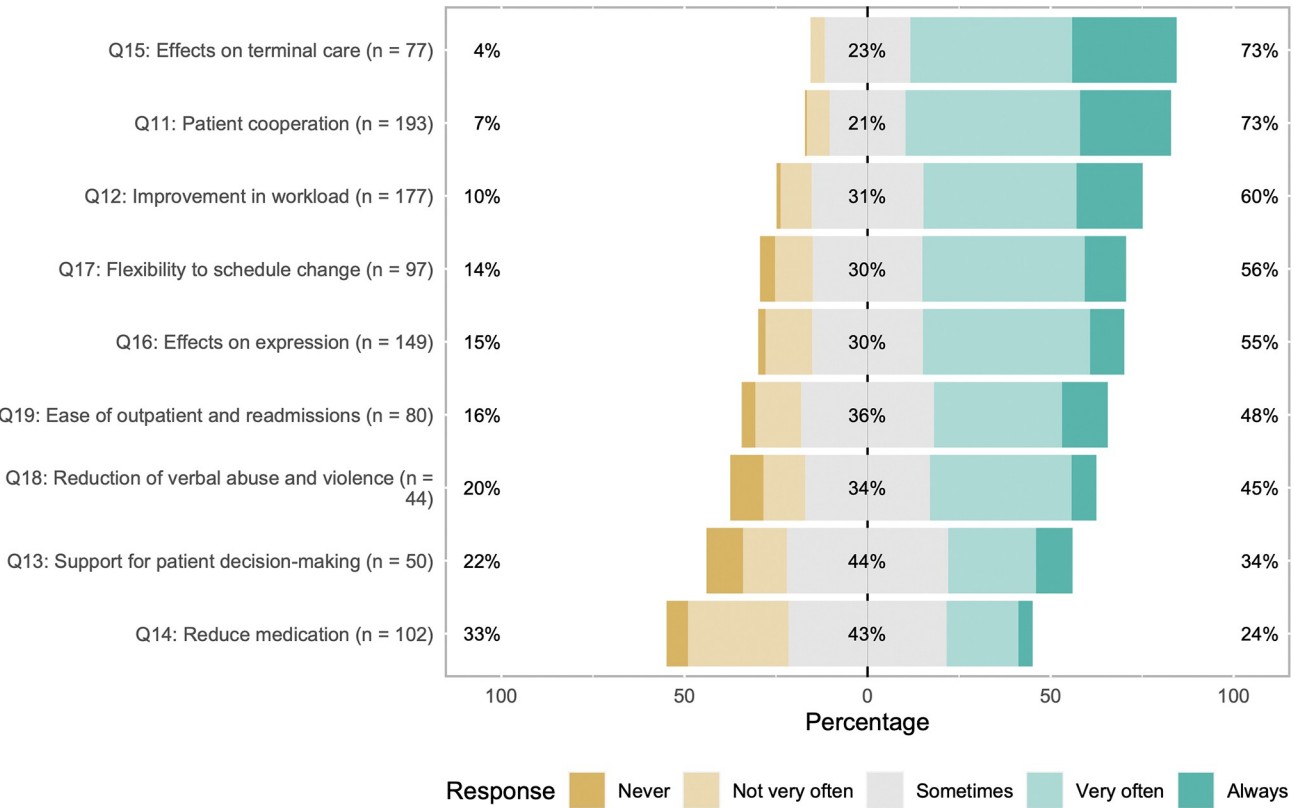

**Fig 2. Visualization of the 270 participants' responses by the frequency of experiencing the impact of facility dog intervention.** This figure is a stacked bar graph created using the likert package in R [27], with the median of the "sometimes" category in the middle and extending to both sides. The numbers on the left side of the bar graph indicate the percentage of healthcare professionals who responded with "Never" or "Not very often." The number in the center of the bar graph indicates the percentage of healthcare professionals who responded with "Sometimes." The numbers on the right side of the bar graph indicate the percentage of healthcare professionals who responded with "Very often" or "Always".

We conducted the same analysis with data from 431 respondents, including the 161 staff members who had never directly accompanied the HFD's activities and answered "no" to Q8, and confirmed that there were no big differences in the results. The top-ranked item was Q11 "Patient Cooperation," followed by Q15 "Impact on Terminal Care," and the lowest-ranked item was Q14 "Reducing medication such as sedatives during examinations or treatments." (S1 Fig and S1 Table)

## Open-end description of specific situations regarding the impact of HFD activities

For Q15, "Impact on terminal care," 23 of the 270 respondents (9%) provided open-ended responses in writing (Table 4). The responses were obtained from nurses who belonged to the following wards: Internal Medicine children's (age >3 years) ward (6 responses), Cardiac Care Unit, Psychological Treatment for Children and Family (3 responses), Cardiovascular Ward (2 responses), Surgical Department (2 responses), Pediatric Intensive Care Unit (1 response), and Neonatal Intensive Care Unit and Growing Care Unit (1 response).

Regarding Q12, "Improvement in workload," open-ended responses were obtained from 65 of 177 respondents (56%) (S2 Table). The responses from the Internal medicine children's ward (age >3 years) included specific examples of procedures such as bone marrow puncture, lumbar puncture, taking patients to the operating room, and securing the peripheral intravenous lines.

**Table 3. Comparison of the results by respondent attributes for each of the nine questions.**

| | | Profession | | | | | Clinical experiences | | | | | Experience having a dog | | | | |
|---|---|---|---|---|---|---|---|---|---|---|---|---|---|---|---|---|
| | | $\chi^2$ | df | $p$ | Effect Size | [95% CI] | $\chi^2$ | df | $p$ | Effect Size | [95% CI] | $\chi^2$ | df | $p$ | Effect Size | [95% CI] |
| Q11 | Patient cooperation | 2.67 | 2 | .264 | .08 | [-.06, .22] | 1.86 | 2 | .394 | .06 | [-.08, .20] | 0.35 | 1 | .555 | .04 | [-.10, .18] |
| Q12 | Improvement in workload | 1.00 | 2 | .608 | .04 | [-.11, .18] | 4.26 | 2 | .119 | .12 | [-.03, .26] | 0.23 | 1 | .629 | .04 | [-.11, .18] |
| Q13 | Support for patient decision-making | 2.30 | 2 | .317 | .14 | [-.14, .40] | 4.53 | 2 | .104 | .23 | [-.05, .48] | 1.13 | 1 | .287 | .15 | [-.13, .41] |
| *[1,2]Q14 | Reduce medication | 0.22 | 2 | .898 | .01 | [-.18, .21] | 13.93 | 2 | < .001 | .33 | [.14, .49] | 4.15 | 1 | .042 | .20 | [.01, .38] |
| Q15 | Effects on terminal care | 0.66 | 2 | .719 | .04 | [-.18, .26] | 2.90 | 2 | .235 | .14 | [-.09, .35] | 0.00 | 1 | .981 | .00 | [-.22, .23] |
| Q16 | Effects on expression | 2.57 | 2 | .277 | .09 | [-.07, .25] | 1.54 | 2 | .462 | .06 | [-.10, .22] | 1.52 | 1 | .218 | .10 | [-.06, .26] |
| Q17 | Flexibility to schedule change | 0.91 | 2 | .634 | .05 | [-.15, .25] | 3.01 | 2 | .222 | .12 | [-.08, .32] | 1.34 | 1 | .246 | .12 | [-.08, .31] |
| Q18 | Reduction in verbal abuse and violence | 0.13 | 2 | .939 | .01 | [-.29, .31] | 1.50 | 2 | .472 | .11 | [-.19, .39] | 0.10 | 1 | .754 | .05 | [-.25, .34] |
| *[2]Q19 | Ease of outpatient and readmissions | 3.14 | 2 | .208 | .14 | [-.08, .35] | 0.39 | 2 | .822 | .03 | [-.20, .24] | 4.60 | 1 | .032 | .24 | [.02, .44] |

*$p < .05$. Accepted criteria for effect size $r$: 0.1 (small), 0.3 (medium), 0.5 (large).

*[1]The results of multiple comparison-corrected pairwise comparisons (Mann-Whitney U test) showed that the group with experience of 5 to 20 years ($U = 265.50$, $p = 0.004$, $r$ [95% CI] = 0 0.23 [0.09, 0.36]) and the group with more than 20 years of experience ($U = 127.00$, $p = 0.004$, $r$ [95% CI] = 0.30 [0.14, 0.46]) both had higher scores than the group with less than 5 years of clinical experience. There was no difference between the groups with experience of 5 to 20 years and more than 20 years ($U = 692.00$, $p = 0.885$, $r$ [95% CI] = 0.07 [-0.06, 0.20]).

*[2]It was higher in the group that had an experience of having dog

Open-ended responses to Q14, "Reduced medication," were provided by 18 of 102 respondents (18%) (S3 Table). Specific situations were cited, such as a decrease in the bolus of Patient Controlled Analgesia and non-use of premedication.

## Discussion

In this study, the two highest-rated items were Q15 "Impact on terminal care," and Q11 "Patient cooperation," and the lowest rated item was Q14 "Reduced medication."

One of the situations in which the medical staff found FD activities particularly useful was the intervention in palliative care toward the end of the patient's life (Q15). There were no differences in the attributes of the respondents, indicating that the respondents found the FD activities beneficial, irrespective of their medical profession, years of experience, or past experience of dog ownership. Although the actual number of intervention cases was not counted in this study, the number of death discharges at the hospital was 40 (0.73%) in FY2018 and 27 (0.50%) in FY2019 [28]; therefore, there were not many opportunities for HFD activities and interventions. Yet, the predominantly positive responses suggest that the intervention had a strong impact on terminal care. Presumably, one factor that led to timely interventions despite the limited opportunities was the system that allowed handler to work with various departments within the hospital, enabling the handler to make decisions about patient interventions more flexibly, regarding both the order and frequency of the intervention, based on the gathered information and assessments of other clinical staff. These unique characteristics of HFD teams were identified in this study and have not been found in previous studies on the effects of therapy dogs.

**Table 4. All comments made in the free text field of Q15 "Effects on terminal care".**

| Ward | Profession, Comments | |
|---|---|---|
| Internal medicine children's ward (age >3 years) | | |
| | Ns. | It made the family smile. |
| | Ns. | It provided comfort to our soul. |
| | Ns. | I saw a patient's facial expression soften. |
| | Ns. | It brought smiles on faces. |
| | Ns. | I felt that the entire family had a peaceful time. |
| | Dr. | For a patient with terminal cancer, I felt the facility dog was helpful. |
| Cardiac Care Unit (CCU) | | |
| | Ns. | There was a time when I felt that the family's expression softened when they saw the facility dog. The patient was almost unconscious at the time. |
| | Ns. | When I saw a patient who was looking forward to seeing the facility dog, I thought it was good to have some healing and fun in the midst of painful treatment. |
| | Dr. | It seemed to have given the patient and family members a break from the treatment and pain and provided a relaxing time together for a moment. |
| Psychological Treatment for Children and Family | | |
| | Dr. | I found it (HFD intervention) useful through the palliative care conferences I used to attend. |
| | Dr. | Not only the patient but also the family could spend peaceful time with the facility dog. |
| Cardiovascular Ward | | |
| | Dr. | I feel that the intervention of the facility dog eases the tension of the place somewhat. |
| | Ns. | Both patients and parents can spend a peaceful time together with Yogi. |
| Surgical Department | | |
| | Ns. | It provided support for the whole family. |
| | Ns. | I saw smiles during the interventions. |
| Pediatric Intensive Care Unit (PICU) | | |
| | Ns. | It brought a smile on a patient's face who was in the terminal stage of leukemia. |
| Neonatal Intensive Care Unit (NICU), Growing Care Unit (GCU) | | |
| | Ns. | I have experienced and heard comments from patient's families that they are having a good time just by being around Yogi. Even if the patient could no longer recognize Yogi because of the terminal condition, they could still stay close by and touch him. I would be grateful if Yogi could visit the family in the same way. |
| Others | | |
| | Dr. | The intervention before a kidney transplant surgery made them more positive. |
| | Others | Although it could only be seen through the glass window, the intervention brought smiles on the faces of the parents and the child who was in the terminal stage and created a warm atmosphere. The tension in the room was somewhat eased. |
| | Others | At the visit of the facility dog, there was a calmness like a friend's visit that put both the patient and family at ease. |
| | Others | I saw a patient asking for the facility dog several times. |
| | Others | I have not seen the facility dog team in action in terminal care, but I have heard about it and found it useful. |
| | Others | I have never been in such a situation, but I think that looking forward to seeing the facility dog and being happy to see him can improve the child's QOL (especially in terminal palliative care) and will also affect pain management. If there is an opportunity to learn more, I would love to hear about it. |
| | Others | I thought that a patient who has a dog at home and who loves Yogi would feel safe by touching him even in their final days. |
| | Others | The dog was about to stop in front of the hospital room of a patient who was entering the terminal stage. |

QOL, quality of life.

The results also highlight the fact that timely palliative care can be provided with the help of HFD activities, consistent with the finding reported by Contro and Sourkes [23]. Palliative care is most effective when it is introduced at an early stage. In fact, our program was a full-time program, and in addition to animal-assisted therapy, which provided support for treatment and procedures, we provided animal-assisted activities on a regular basis for patients to enjoy communication with animals. This strategy made it possible to provide palliative care smoothly at the earliest stage possible. In a report from a comprehensive cancer center on the timing of palliative care [29], which compared patients who requested palliative care more than 6 months before death to those who initiated it within 6 months of death, the former (earlier initiation of palliative care) was associated with lower rates of subsequent emergency room visits and hospitalizations than the latter, resulting in significantly better outcomes in the last 30 days of life. By focusing on the timing of palliative care with the help of HFDs, future studies may lead to alternative conceptual models to support timely and targeted interventions.

Importantly, doctors mentioned in their open-ended response that they "felt that (HFD interventions) was useful through palliative care conferences." Palliative care conferences are intended to involve necessary professionals and provide a higher level of palliative care to patients, and handlers are also allowed to participate as professionals. The comment meant that a certain evaluation was obtained among medical professionals at the conference. In Japan, it has been reported that the most sought-after supportive measure by parents in the terminal stage of their child's life is "daily visits to the hospital room to talk to the child (90.2%) [30]." In a study on palliative care centers that analyzed the content of the conversations between handlers and patients, 10.5% were related to the current health status along with death and dying [31]. Because nurses have advantages in physical and informational support compared to other professions and volunteers in the interdisciplinary palliative care team [32], it was inferred that nurse-handlers were effective in addressing supportive care needs, especially at the end of life. Thus, it is expected that handlers may be able to provide better care in the future through the acquisition of advance care planning and other skills as nurses and share information obtained through HFD activities with interdisciplinary palliative care teams to strengthen collaboration, thereby improving the value of terminal medical care with HFDs.

Of the complementary therapies provided by hospice care providers at the end of a patient's life, pet therapy accounts for 58.6% and is the fourth most common after massage and other therapies [33]. The facts that cancer patients perceive an improvement in their symptoms when visited by therapy dogs [34] and pediatric patients experience pain relief from therapy dog interventions [12] were independent of the patients' own dog ownership experience. Interestingly, dog ownership experience on the part of medical professionals was also unrelated to the evaluation of HFD activities. Thus, whether the respondent owned a dog in the past did not influence their response to this item, indicating that the result was bias-free and generalizable.

As the open-ended response to Question 15 had many descriptions of positive emotions, especially the word, "smiles," it can be suggested that HFD activities can bring about more happiness than conventional methods to patients, their parents, and caregivers, even in the terminal stage of the patient's life, corroborating previous findings on the use of pet therapy in terminal care [35].

"Patient cooperation" (Q11) also received high scores in this study probably owing to the fact that the nurse-handlers were skilled in infection control, had completed training according to the international guidelines for the handling of dogs, and were thus able to engage safely in examinations and procedures that require cleanliness. In contrast, a large-scale survey of therapy dog organizations revealed that only a small number of organizations prohibited "feeding raw meat meals or treats to dogs" that pose a risk of zoonotic disease transmission

[36] and 20% of volunteer handlers did not follow infection control measures [37]. In the current study, many situations that require higher levels of infection control and have not been reported in previous studies were evaluated in the open-ended responses, such as (a) bone marrow and lumbar puncture and (b) taking patients to the medical operating room (S2 Table). Therefore, nurse-handlers can lead and support the development of high quality animal assisted therapy programs in hospitals [13].

"Reduced medication" (Q14) was reported as the item with the lowest evaluation. It was the only item among all nine items in this study for which the percentage of negative evaluations exceeded that of positive ones. This finding is in contradiction to that of previous studies that reported a reduction in postoperative analgesic doses in adults after several visits from therapy dogs [22], a reduction in sedative use by residents before and after six months of resident dog activities in rehabilitation facilities [14], and improved analgesic effects in children [13]. Moreover, statistical differences were observed among the respondents in terms of the years of clinical experience and experience of dog ownership. Respondents with longer clinical experience or dog ownership experiences rated it higher. The latter result may well have been affected by bias. In the past, "generalized pet effects" on health and well-being have demonstrated inconsistent results [38]. For example, a study found that dog owners rated daily stressors more strongly than non-owners, leading the authors to argue that the dog's protective role regarding stress reduction is overestimated in the media and research [39–40]. However, on the other hand, it is considered that one factor contributing to the lowest ratings is the influence of the hospital or attending physician's prescription protocol. For example, in this hospital, even for sedatives, a policy is taken to "use them as little as possible in cases where sedatives can be avoided," regardless of the presence of a facility dog. In the free description section, it is also pointed out in a comment that "I do not think that the amount of medication has changed since the child did not need premedication," suggesting that the medication may have been reduced according to the original prescription protocol. This issue needs further attention in future studies by strictly eliminating the confounding factors and distinguish drug types from sedatives and analgesics etc.

## Study limitations

Recall bias may have existed because this study relied on the retrospective evaluation by respondents. Only two handler–dog pairs were involved in the study, and thus, it may not be possible to generalize the results with regard to the impact of the activities of nurse-handlers and HFDs. The results were obtained from a small-scale children's hospital with 274 beds. Further studies in more facilities are necessary to generalize the findings.

## Clinical implication

A full-time operation model with HFDs and nurse handlers could be useful in supporting patients in children's hospitals. Outcomes may be commensurate with the costs. An evaluation method called Social Return On Investment (SROI), which converts social impact into monetary value, should be considered. Further research is needed to devise timely and targeted interventions and to identify optimal operational methods to meet the needs of the HFD implementing facilities.

## Conclusions

We investigated situations in which the hospital medical staff are likely to feel the impact of full-time activities of HFDs and nurse-handlers. The results revealed their usefulness in end-of-life interventions and the ease with which patients cooperate with highly invasive

procedures. To maximize the efficacy of limited resources, healthcare providers and hospital administrators need to be aware of these characteristics when making decisions to utilize HFD programs.

## Supporting information

**S1 Fig. Visualization of the 431 participants' responses based on the frequency of experiencing the impact of facility dog intervention.** This figure is a stacked bar graph created using the likert package in R [27], with the median of the "sometimes" category in the middle and extending to both sides. The numbers on the left side of the bar graph indicate the percentage of healthcare professionals who responded with ""Never" or "Not very often". The number in the center of the bar graph indicates the percentage of healthcare professionals who responded with "Sometimes". The numbers on the right side of the bar graph indicate the percentage of healthcare professionals who responded with "Very often" or "Always".
(DOCX)

**S1 Table. Comparison of the results of nine questions (Q11-Q19) based on the attributes of the 431 respondents to Q8 "Frequency of accompaniment to interventions".**
(DOCX)

**S2 Table. All comments made in the free text field of Q12 "Improvement in workload".**
(DOCX)

**S3 Table. All comments made in the free text field of Q14 "Reduced medication".**
(DOCX)

## Acknowledgments

We would like to thank ACCEA Co., Ltd., for printing support, Mariko Yamamoto, Ph.D., Sayaka Kuze, Ph.D., Dr. Tomoko Takayanagi for constructive feedback to improve the quality of the paper. We also thank the Assistance Dogs of Hawaii for allowing their wonderful facility dogs to come to Japan and Tomoko Miyoshi and Yayoi Suzuki for assisting with data compilation and writing. Last but not least, we would like to thank staff members at Shizuoka Children's Hospital for their cooperation in answering our questions and the supporters of Shine On! Kids, a certified NPO. This paper is dedicated to Tyler (aged 1), who gave us the inspiration to establish Shine On! Kids, and to the late facility dog Bailey, who passed away in 2020.

## Author Contributions

**Conceptualization:** Natsuko Murata-Kobayashi, Shiro Seto.

**Data curation:** Natsuko Murata-Kobayashi, Keiko Suzuki, Yuko Morita, Harumi Minobe.

**Formal analysis:** Natsuko Murata-Kobayashi, Atsushi Mizumoto.

**Investigation:** Natsuko Murata-Kobayashi, Keiko Suzuki, Atsushi Mizumoto, Shiro Seto.

**Methodology:** Natsuko Murata-Kobayashi, Atsushi Mizumoto, Shiro Seto.

**Project administration:** Natsuko Murata-Kobayashi, Shiro Seto.

**Resources:** Natsuko Murata-Kobayashi, Keiko Suzuki, Yuko Morita, Harumi Minobe, Shiro Seto.

**Software:** Natsuko Murata-Kobayashi, Atsushi Mizumoto.

**Supervision:** Natsuko Murata-Kobayashi, Atsushi Mizumoto, Shiro Seto.

**Validation:** Natsuko Murata-Kobayashi, Atsushi Mizumoto.

**Visualization:** Natsuko Murata-Kobayashi, Atsushi Mizumoto, Shiro Seto.

**Writing – original draft:** Natsuko Murata-Kobayashi.

**Writing – review & editing:** Natsuko Murata-Kobayashi, Atsushi Mizumoto, Shiro Seto.

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
