## [Decision Letter · Decision Letter 0]

6 Feb 2023

PONE-D-22-26487Exploring the benefits of full-time hospital facility dogs working with nurse handlers in a children’s hospital.PLOS ONE

Dear Dr. Murata,

Thank you for submitting your manuscript to PLOS ONE. After careful consideration, we feel that it has merit but does not fully meet PLOS ONE’s publication criteria as it currently stands. Therefore, we invite you to submit a revised version of the manuscript that addresses the points raised during the review process.

We look forward to receiving your revised manuscript.

Kind regards,

Krit Pongpirul, MD, MPH, PhD.

Academic Editor

PLOS ONE

Journal Requirements

4. Please include your tables as part of your main manuscript and remove the individual files. Please note that supplementary tables (should remain/ be uploaded) as separate "supporting information" files

Additional Editor Comments:

Please address all of the comments and suggestions from both reviewers. If possible, please include/integrate responses from the caregivers who did not have direct experiences with HFD.

Reviewers' comments:

Reviewer's Responses to Questions

**Comments to the Author**

1. Is the manuscript technically sound, and do the data support the conclusions?

Reviewer #1: Yes

Reviewer #2: Yes

2. Has the statistical analysis been performed appropriately and rigorously? 

Reviewer #1: Yes

Reviewer #2: Yes

3. Have the authors made all data underlying the findings in their manuscript fully available?

Reviewer #1: No

Reviewer #2: Yes

4. Is the manuscript presented in an intelligible fashion and written in standard English?

Reviewer #1: Yes

Reviewer #2: Yes

5. Review Comments to the Author

Reviewer #1: The article reports a questionnaire survey of caregivers in a pediatric hospital on their opinions about the effects of hospital facility dogs (HFDs) working with qualified nurse handlers in various care situations.

The analysis includes 270 responses from caregivers who had at least one experience with a care procedure in the presence of a HFD.

The most positive effects are observed in palliative care situations and for patient cooperation in care. The least important effect is related to the reduction of medication.

The topic is very original and interesting. It is a declarative survey with a low level of evidence. Nevertheless, the results show that the contact of children's patients with HFDs can improve care relationships and have a place among complementary therapies. These results open up directions for reflection and hypotheses to be explored.

The article is clear, well-written, and there are no formatting issues.

Note: I find it unfortunate that the authors excluded responses from caregivers who have not had direct personal experience with HFDs. These caregivers working in the same hospital must also have an opinion about HFDs. It would have been interesting to compare their opinions with those of caregivers who have had experience with HFDs.

Reviewer #2: A valuable intervention and a good paper describing its potential value. .

It is however not clear why only half of the questionnaire questions are reported here.. are the others described somewhere else?

I also have some specific queries/ suggestions

Page 2 paragraph 1 where do HFDs live? You mention they are different to therapy dogs which are pets, which implies these are not pets

Page 2 final paragraph “studies on volunteer therapy” should read “studies OF volunteer therapy”

Patients with allergies or fears of dogs were excluded from use… can an estimate be given of what proportion of patients this constitutes?

Page 4 final paragraph .. why was service years categorised? I fear this grouping may have hampered your ability to detect subtle effects of time of service

Page 6 line I don’t understand how a mean time of service was calculated if the questions was categorised as mentioned above. if the variable is categorical isn’t a model category more representative?

Fig 2 I am not familiar with the sloping presentation of this stacked bar chart, please expand the legend to explain how each bar is centred

Page 6 Section 3.2 Why are the numbers answering each item so low e.g. Of the 270 respondents, 77 (29%) answered this item, o. why did respondents miss items?

Table 4-6 are quite long.. could the answers be thematically organised to make them easier to read?

The questions are also worded such that they encourage positive responses, was any attempt made to find out whether some thought the dog for example increased workload?

Page 7 Discussion The first sentence needs to remind the reader what each question referred do was

Page 8 paragraph 2 It is not clear what is meant by “felt that (HFD intervention) was useful through palliative care conferences.” - please elaborate

Page 9 final paragraph the effect of factors on Q14 possibly due to the hospital’s protocol? Are there prescribed guidelines on dosages and hence most medical practitioners would err on the side of caution and not reduce this in the dogs presence?

Page 10 paragraph 2 The statement “Outcomes may be commensurate with the costs seems out of the blue.. can this be fully justified? Can any analysis to be included or suggested for the future to fully support this claim?

6. PLOS authors have the option to publish the peer review history of their article (what does this mean?). If published, this will include your full peer review and any attached files.

Reviewer #1: **Yes: **Patrice François

Reviewer #2: **Yes: **Nicola Rooney

---

## [Author Response · Author response to Decision Letter 0]

13 Mar 2023

Academic Editor’s Comments

Academic Editor’s Comment 1

Thank you for submitting your manuscript to PLOS ONE. After careful consideration, we feel that it has merit but does not fully meet PLOS ONE’s publication criteria as it currently stands. Therefore, we invite you to submit a revised version of the manuscript that addresses the points raised during the review process.

Author Response:

Thank you very much for your constructive feedback. It feels as if you and the two reviewers were my co-authors. We have worked on this major revision of the paper to address all of the concerns and comments raised by the reviewers. We hope the manuscript now has additional breadth and depth as you expect.

Academic Editor’s Comment 2

Please address all of the comments and suggestions from both reviewers. If possible, please include/integrate responses from the caregivers who did not have direct experiences with HFD.

Author Response:

Thank you very much for your important feedback. We noted that certain respondents who were not included in the analysis had never accompanied a facility dog to a location where it provided intervention. Specifically, medical professionals who had only encountered a facility dog in the hospital corridor may have responded with bias due to their imagination. As such, this should be treated as a confounding factor. After careful consultation with the two co-author handlers (KS and YM), who have a good understanding of the actual situation in the field, we decided to exclude these respondents from the survey. 

Additionally, we confirmed that the results were not significantly different when we added staff members who had never directly observed the facility dog to the same analysis. We revised the manuscript (p22) and included the details of the analysis in the Online Supplementary Material.

Reviewer 1’s Comments

Reviewer 1’s Comment 1

Reviewer #1: The article reports a questionnaire survey of caregivers in a pediatric hospital on their opinions about the effects of hospital facility dogs (HFDs) working with qualified nurse handlers in various care situations.

The analysis includes 270 responses from caregivers who had at least one experience with a care procedure in the presence of a HFD.

The most positive effects are observed in palliative care situations and for patient cooperation in care. The least important effect is related to the reduction of medication.

The topic is very original and interesting. It is a declarative survey with a low level of evidence. Nevertheless, the results show that the contact of children's patients with HFDs can improve care relationships and have a place among complementary therapies. These results open up directions for reflection and hypotheses to be explored.

The article is clear, well-written, and there are no formatting issues.

Note: I find it unfortunate that the authors excluded responses from caregivers who have not had direct personal experience with HFDs. These caregivers working in the same hospital must also have an opinion about HFDs. It would have been interesting to compare their opinions with those of caregivers who have had experience with HFDs.

Author Response:

Thank you for taking the time to read and provide feedback on our article. We appreciate your positive comments on our manuscript. We have addressed all the issues and concerns and other details suggested by you and the other reviewer. We believe that, by addressing all comments, the manuscript is now clearer and more appealing than it was before. 

As for your note (i.e., excluding responses from caregivers who have not had direct personal experience with HFDs), certain respondents who were not included in the analysis had never accompanied a facility dog to a location where it provided intervention. Specifically, medical professionals who had only encountered a facility dog in the hospital corridor may have responded with bias due to their imagination. As such, this should be treated as a confounding factor. After careful consultation with the two co-author handlers (KS and YM), who have a good understanding of the actual situation in the field, we decided to exclude these respondents from the survey. 

Additionally, we confirmed that the results were not significantly different when we added staff members who had never directly observed the facility dog to the same analysis. We revised the manuscript (p22) and included the details of the analysis in the Online Supplementary Material.

Reviewer 2’s Comments

Reviewer 2’s Comment 1

A valuable intervention and a good paper describing its potential value. It is however not clear why only half of the questionnaire questions are reported here.. are the others described somewhere else?

Author Response:

Thank you for your feedback. Our questionnaire was specifically designed to investigate the effects of interventions that handlers and facility dogs typically engage in, focusing on items from Q11 and beyond. As the items from Q1 to Q10 were not directly related to our research objective, we provided information on these items exclusively in the Online Supplementary Material and noted this in the main text (p9).

Reviewer 2’s Comment 2

Page 2 paragraph 1 where do HFDs live? You mention they are different to therapy dogs which are pets, which implies these are not pets

Author Response:

Thank you for your question. As it is a valid point, we have included the following explanation in the main text (p3):

HFDs differ from therapy dogs, which are trained pets [1], in that they are as professionally trained as guide dogs and service dogs. HFDs also usually live with their handlers as caretakers and commute with them every morning.

Reviewer 2’s Comment 3

Page 2 final paragraph “studies on volunteer therapy” should read “studies OF volunteer therapy”

Author Response:

Thank you for the suggestion. We have changed “on” to “of.”

Reviewer 2’s Comment 4

Patients with allergies or fears of dogs were excluded from use… can an estimate be given of what proportion of patients this constitutes?

Author Response:

Thank you for your question. It is extremely rare for a visit to be withheld because of allergies or a fears of dogs, and we estimate that less than 10% of eligible patients did not wish to be visited, which is small. We have included this explanation in the text (p8). 

Reviewer 2’s Comment 5

Page 4 final paragraph .. why was service years categorised? I fear this grouping may have hampered your ability to detect subtle effects of time of service

Author Response:

Thank you for your suggestion. We used quartiles for grouping to explore the impact of years of clinical experience, as no previous studies have addressed shorter or longer years of clinical experience, and this was an exploratory study. In response to your suggestion, we have also added the reasons for categorizing by years of clinical experience in the manuscript along with the cited references (p9).

Reviewer 2’s Comment 6

Page 6 line I don’t understand how a mean time of service was calculated if the questions was categorised as mentioned above. if the variable is categorical isn’t a model category more representative?

Author Response:

Thank you for bringing this to our attention. In this section, we presented the data as descriptive statistics to aid readers in understanding the distribution of the data. In the revised manuscript, we have added the median in addition to the mean (p17). The raw data are also available as supplementary material on the Open Science Framework (https://osf.io/hxktc/?view_only=aa2725be0afb479ca67a530421e32848), allowing readers to refer to them and conduct their own calculations when similar questions arise.

Reviewer 2’s Comment 7

Fig 2 I am not familiar with the sloping presentation of this stacked bar chart, please expand the legend to explain how each bar is centred

Author Response:

Thank you for your feedback. We have added an explanation of how to read the graph to the legend.

Reviewer 2’s Comment 8

Page 6 Section 3.2 Why are the numbers answering each item so low e.g. Of the 270 respondents, 77 (29%) answered this item, o. why did respondents miss items?

Author Response:

Thank you for your inquiry. The reason for the low number of responses is related to the frequency of the interventions we assumed for each item. For instance, concerning terminal care (Q15), the number of deaths discharged from the same hospital is 27 per year (in 2019). Therefore, even if HFD intervenes in all 27 cases, the staff present at the scene is very limited. In contrast, situations where patient cooperation can be obtained (Q11) occur frequently in daily life, and there are many opportunities to witness such scenes. As a result, we received responses from 193 people (71%). 

We have included this explanation in the text (p21).

Reviewer 2’s Comment 9

Table 4-6 are quite long.. could the answers be thematically organised to make them easier to read?

Author Response:

Thank you for your suggestion. We have focused on “Terminal Care” in Table 4 only and moved Tables 5 and 6 to Online Supplementary Material.

Reviewer 2’s Comment 10

The questions are also worded such that they encourage positive responses, was any attempt made to find out whether some thought the dog for example increased workload?

Author Response:

Thank you for your question. In items 4-10 of the questionnaire used in this paper, we asked, for example, “Q10 Have you ever felt that the intervention of the Facility Dog was a burden on your work?” and other questions (Table 1). Note that the results indicated that few respondents felt that workload increased. In this paper, we focused on the analysis of items after Q11 with the aim of advancing to quantitative research with a higher level of evidence in the future. Q1–Q10 are not directly related to the purpose of the study and are reported in Online Supplementary Material.

For items after Q11, we asked questions that focus on interventions that handlers can be more effectively involved in on a daily basis and attempted to determine which of these interventions would also be evaluated from an objective standpoint by the medical staff. Therefore, the wording of the questions did not necessarily encourage positive responses. In fact, we received honest comments such as “I think they provide psychological support, but it is not connected in terms of work facilitation, and I think it is difficult to connect” in the free response to Q12 regarding improving workload. We believe that we made efforts to reduce the bias that leads to under-reporting of undesirable responses.

Reviewer 2’s Comment 11

Page 7 Discussion The first sentence needs to remind the reader what each question referred do was

Author Response:

Thank you for the suggestion to increase readability. We have made the necessary correction (p25).

Reviewer 2’s Comment 12

Page 8 paragraph 2 It is not clear what is meant by “felt that (HFD intervention) was useful through palliative care conferences.” - please elaborate

Author Response:

Thank you for your question. This phrase means that a doctor who has accompanied HFD interventions has not directly experienced terminal care but has participated in palliative care conferences, obtained information sharing about the actual HFD patient interventions and patient family reactions, and felt that HFD interventions were useful.

Palliative care conferences are intended to involve necessary professionals and provide a higher level of palliative care to patients, and handlers are also allowed to participate as professionals. Since the comment meant that a certain evaluation was obtained among medical professionals at the conference, we quoted it in the text. We have added this description of the palliative care conference to the text (p27) and the following supplemental description to the table (p23):

“I found it (HFD intervention) is useful through the palliative care conferences I used to attend.”

Reviewer 2’s Comment 13

Page 9 final paragraph the effect of factors on Q14 possibly due to the hospital’s protocol? Are there prescribed guidelines on dosages and hence most medical practitioners would err on the side of caution and not reduce this in the dogs presence?

Author Response:

Thank you for your insightful comment. As you point out, it is also believed that the hospital protocol is a factor that affects Q14. We have added it to our manuscript (p30–31).

Reviewer 2’s Comment 14

Page 10 paragraph 2 The statement “Outcomes may be commensurate with the costs seems out of the blue.. can this be fully justified? Can any analysis to be included or suggested for the future to fully support this claim?

Author Response:

Thank you for your comment. Although we do not have data that can be incorporated into the analysis in this paper, we are planning to propose a method of evaluation called social return on investment (SROI) in future research, which involves converting social impact into monetary value. We have added the above supplement to the manuscript (p32).

---

## [Decision Letter · Decision Letter 1]

2 May 2023

Exploring the benefits of full-time hospital facility dogs working with nurse handlers in a children’s hospital.

PONE-D-22-26487R1

Dear Dr. Murata,

We’re pleased to inform you that your manuscript has been judged scientifically suitable for publication and will be formally accepted for publication once it meets all outstanding technical requirements.

Kind regards,

Krit Pongpirul, MD, MPH, PhD.

Academic Editor

PLOS ONE

Additional Editor Comments (optional):

Yours responses to the comments from both reviewers are satisfactory.

Reviewers' comments:

Reviewer's Responses to Questions

**Comments to the Author**

1. If the authors have adequately addressed your comments raised in a previous round of review and you feel that this manuscript is now acceptable for publication, you may indicate that here to bypass the “Comments to the Author” section, enter your conflict of interest statement in the “Confidential to Editor” section, and submit your "Accept" recommendation.

Reviewer #2: All comments have been addressed

2. Is the manuscript technically sound, and do the data support the conclusions?

Reviewer #2: Yes

3. Has the statistical analysis been performed appropriately and rigorously? 

Reviewer #2: (No Response)

4. Have the authors made all data underlying the findings in their manuscript fully available?

Reviewer #2: Yes

5. Is the manuscript presented in an intelligible fashion and written in standard English?

Reviewer #2: Yes

6. Review Comments to the Author

Reviewer #2: Thank you for addressing our comments so diligently. This is a valuable piece of work and I look forward to seeing it in print

7. PLOS authors have the option to publish the peer review history of their article (what does this mean?). If published, this will include your full peer review and any attached files.

Reviewer #2: **Yes: **Nicola Jane Rooney

---

## [Editor Report · Acceptance letter]

9 May 2023

PONE-D-22-26487R1 

Exploring the benefits of full-time hospital facility dogs working with nurse handlers in a children’s hospital 

Dear Dr. Murata-Kobayashi:

I'm pleased to inform you that your manuscript has been deemed suitable for publication in PLOS ONE. Congratulations! Your manuscript is now with our production department. 

Kind regards, 

on behalf of

Assoc. Prof. Dr. Krit Pongpirul 

Academic Editor

PLOS ONE